# Eph receptor tyrosine kinases are functional entry receptors for murine gammaherpesvirus 68

**Anna K. Großkopf**[1]*, **Victor Tobiasson**[2], **Laurie T. Krug**[1]

**1** HIV and AIDS Malignancy Branch, Center for Cancer Research, National Cancer Institute, Bethesda, Maryland, United States of America, **2** Computational Biology Branch, National Library of Medicine, Bethesda, Maryland, United States of America

* anna.grosskopf@nih.gov

## Abstract

Interactions between viral glycoproteins and cellular receptors determine virus tropism and represent promising targets for vaccines. Eph receptor tyrosine kinases are conserved receptors for the human oncogenic gammaherpesviruses, Kaposi sarcoma herpesvirus (KSHV) and Epstein-Barr virus (EBV), and mediate entry into target cells by interaction with the viral gH/gL glycoprotein complex. To evaluate the use of murine gammaherpesvirus 68 (MHV68), a natural pathogen of rodents, as an *in vivo* model system for early events in gammaherpesvirus infection, we characterized the interaction of the MHV68 gH/gL complex with Eph receptors. We demonstrate a direct interaction of MHV68 gH/gL with EphA4 and EphB3, that is conserved between human and murine receptors. Pre-incubation of MHV68 inocula with soluble decoy receptors decreased infection of permissive fibroblasts. Ectopic expression of EphA4 and EphB3 enabled MHV68 to infect otherwise non-permissive human B cells, demonstrating EphA4 and EphB3 receptor function. Targeted mutations informed by protein structure predictions demonstrate that the MHV68 gH/gL-Eph interaction is determined by domain I (D-I) and follows structural motifs previously described in the KSHV gH/gL-EphA2 complex. The importance of gH D-I is further highlighted by the analysis of gH-targeting neutralizing antibodies. Antibody adsorption via the full gH ectodomain or gH D-I led to comparable reductions in neutralization capacity of serum from WT infected mice, indicating the Eph-binding domain is a major target for gH/gL-directed neutralizing antibodies. Our study characterizes Eph receptors as novel interaction partners and entry receptors for MHV68. Conservation of entry mechanisms provides the basis for future *in vivo* analyses of the contribution of Eph receptors to cell-type dependent MHV68 infection, as well as targeted strategies to prevent transmission and diseases associated with chronic infection.

the Creative Commons CC0 public domain dedication.

**Data availability statement:** All relevant data are within the manuscript and its Supporting information files.

**Funding:** This research was supported [in part] by the Intramural Research Program of the National Institutes of Health (NIH) and the National Cancer Institute (NCI) ZIA BC 011953 (L.T.K.). A.K.G. was supported by an Intramural AIDS Research Fellowship funded by the NIH Office of AIDS Research and administered by the NIH Office of Intramural Training and Education. The contributions of the NIH author(s) are considered Works of the United States Government. The findings and conclusions presented in this paper are those of the author(s) and do not necessarily reflect the views of the NIH or the U.S. Department of Health and Human Services. The funders had no role in study design, data collection and analysis, decision to publish, or preparation of the manuscript.

**Competing interests:** The authors have declared that no competing interests exist.

## Author summary

Virus-specific intervention and prevention is needed to fight cancers and morbidity caused by the oncoviruses Kaposi sarcoma herpesvirus (KSHV) and Epstein-Barr Virus (EBV). The targeted design of such strategies requires a detailed knowledge of early events in the viral life cycle and the contribution of cellular receptors to infection and spread *in vivo.* Here, we characterize the role of Eph receptors in murine gammaherpesvirus 68 (MHV68) infection, a well-established mouse model for gammaherpesvirus (GHV) pathogenesis and preclinical therapeutic evaluations. As described for human GHVs, the interaction of Eph receptors with the gH/gL complex is mediated by a conserved motif comprised of the N-terminal domain of glycoprotein gH co-folded with gL and contributes to variable levels to infection of different target cells. We demonstrate that antibodies directed against the N-terminal domain of the gH/gL glycoprotein complex are generated upon MHV68 infection in mice and neutralize MHV68 infection in cell culture. Our data enables the analysis of the contribution of Eph receptor engagement to GHV infection *in vivo* and provides a promising target for subunit vaccination approaches.

## Introduction

Despite improved outcomes under anti-retroviral therapies, the risk of cancers is still significantly higher in people living with HIV (PWH) worldwide [1,2]. Major contributors to this elevated risk are oncogenic viruses, specifically the gammaherpesviruses (GHV) Kaposi sarcoma herpesvirus (KSHV, human herpesvirus 8 or *Rhadinovirus humangamma8*) and Epstein-Barr virus (EBV, human herpesvirus 4 or *Lymphocryptovirus humangamma4*) [2,3]. KSHV is the etiological agent of Kaposi sarcoma, a highly vascularized tumor of the skin and mucosal surfaces that is a leading cause of death for PWH in areas of endemic KSHV infection [2,4]. KSHV also drives lymphoproliferative diseases including primary effusion lymphoma (PEL) and a variant of multicentric Castleman disease [5,6]. EBV is linked to a wide range of lymphoproliferative diseases, including Burkitt lymphoma, Hodgkin lymphoma and diffuse large B cell lymphomas [7–9], in addition to epithelial cell tumors including nasopharyngeal carcinoma and gastric carcinoma [7,10,11]. Primary infection with EBV in adolescents and young adults may present as infectious mononucleosis [12] and an etiological association of EBV infection with multiple sclerosis is established [13–15]. Given the substantial global health impact, strategies that prevent infection or reduce viral burden are critical to prevent GHV-associated diseases and cancers.

The wide range of malignancies associated with GHVs derives from their broad cell tropism and reservoirs of infection [16], presenting a key challenge for the development of KSHV and EBV vaccines. Herpesvirus entry into host cells is a multi-step process initiated by attachment to the cell surface, followed by virus uptake. Multiple GHV glycoproteins engage attachment factors in a manner that is largely

independent of target cell type, helping to concentrate the virus on the cell surface. In contrast, subsequent interactions with specific entry receptors, that may vary with cell type, trigger viral uptake and fusion [17,18]. Decoding virus-host interactions that enable entry followed by productive replication or quiescent latency in multiple cellular reservoirs *in vivo* is critical for developing targeted vaccination approaches.

In addition to virus-specific glycoproteins, EBV and KSHV encode envelope glycoproteins that are broadly conserved among herpesviruses, namely the core herpesviral fusion machinery consisting of glycoprotein B (gB) and the gH/gL complex, as well as the gM/gN complex. The GHV gH/gL complex is essential for GHV infection of a broad range of target cells, with a role in both attachment and entry and represents a major target for neutralizing antibody responses in humans and mice [19–23]. On B cells, EBV entry is mediated by the interaction of HLA class II with a heterotrimer consistent of the gH/gL and gp42 [24,25], followed by gB-mediated fusion of viral and host membranes. On the other hand, entry of EBV into epithelial cells is a gp42-independent process facilitated by the interaction of gH/gL heterodimer with non-muscle myosin-IIA (NMHC-IIA) and Eph receptor tyrosine kinase A2 (EphA2) [26–28].

In contrast to the cell type-specific use of different gH/gL complexes exhibited by EBV, KSHV uptake is mediated by the gH/gL heterodimeric complex in all instances analyzed thus far. Initially, EphA2 was described as a high affinity KSHV entry receptor, with additional A-type Eph receptors EphA4, EphA5 and EphA7 providing compensatory functions in the absence of EphA2 expression [26,27,29–32]. Eph-dependent entry via EphB3 and EphA7 has been demonstrated for the rhesus macaque rhadinovirus (RRV), a closely related non-human primate (NHP) pathogen model system [30,33]. In addition, EphA2 has recently been identified as an entry receptor for human cytomegalovirus, a betaherpesvirus [34], indicating broader conservation within the Family *Orthoherpesviridae*.

Ephs represent the largest family of receptor tyrosine kinases, with 14 members in the human genome, classified as either A- or B-type based on their binding affinity to the five GPI-linked ephrin-A ligands or the three transmembrane ephrin-B ligands [35]. While EphA and EphB subtypes preferentially engage ligands of their respective class, EphA4 and EphB2 represent exceptions and exhibit a broader interaction pattern that encompassed both A- and B-type ephrins [36,37]. The Eph-ephrin systems play crucial roles in cellular positioning and organization. This is reflected in their physiological role in tissue patterning, and compartmentalization, for example in neural development, but dysregulation of this finely tuned system impacts cancer development, progression, and metastasis [35,38]. In concordance with the complexity of receptor and ligand expression and subsequent signaling events, multiple Eph receptors can have pro- or anti-tumorigenic roles dependent on the type of cancer, somatic mutations, co-factors as well as expression levels of receptors and ligands [39–46].

Animal models are critical to inform key virus-host interactions as targets for effective blockade of host colonization and viral disease. Murine gammaherpesvirus 68 (MHV68, murid herpesvirus 4 or *Rhadinovirus muridgamma4*) is a natural GHV of rodents that serves as a model pathogen to study GHV infection and pathogenesis *in vivo* [47,48]. MHV68 shares biological features with both human GHVs including tropism for B cells, macrophages, epithelial and endothelial cells, and an association with lymphoproliferative disease in immunocompromised animals. MHV68 is most closely related to KSHV, as evidenced by genome colinearity and direct homologs of 69 of the 82 KSHV genes [49]. We recently reported that a replication dead viral mutant of MHV68 elicits protection against WT challenge in immune competent mice and protection of *Ifnar1*$^{-/-}$ mice from virus-driven lethality [50]. Vaccine-induced protection involves both neutralizing antibody responses and virus-specific CD8 T cell-mediated immunity [51]. The GHV gH/gL complex is a major target of neutralizing antibody (nAb) responses [19–23]. While gL is not essential for MHV68 infection and latency establishment *in vivo* [52], MHV68 gH adopts a gL-dependent conformation that influences viral attachment and entry in a cell type-dependent manner [22,23,52–54]. However, specific cellular receptors used by MHV68 glycoprotein complexes to gain entry have not been characterized.

Here, we evaluated if MHV68 gH/gL interacts with Eph receptors to mediate cell entry. We demonstrate a direct interaction of EphA4 and EphB3, receptors for KSHV and RRV, respectively, with the MHV68 gH/gL complex. Using a combination of

*in silico* modeling, cell culture and *in vivo* experiments, we reveal that MHV68 interactions with Eph receptors parallel those seen in human pathogenic GHV infections. Our findings support MHV68 as a relevant system for investigating the *in vivo* relevance of Eph receptors during GHV infection. Understanding this interaction is crucial for evaluating how vaccine-induced antibodies targeting this conserved virus-host entry pathway affect infection dynamics in specific cell types.

## Results

### MHV68 gH/gL interaction profile with human and murine Eph receptors

EBV and KSHV gH/gL structures demonstrate co-folding of gH D-I and gL that provides the basis for gH/gL-Eph interactions [55,56]. To evaluate if this gL-dependent conformation is conserved in MHV68, we modelled the MHV68 gH ecto-domain (gH^ecto) in complex with gL using alphafold2 multimer (Fig 1A). The predicted MHV68 gH/gL structures were found to be similar to experimental structures of gH/gL from orthologous herpesviruses [57–61]. Based on the EBV gH domain structure, MHV68 gH^ecto was divided into four domains: D-I (V25-P86), D-II (L87-S369), D-III (N370-L552), and D-IV (P553-Q702). The Eph binding interface of KSHV and EBV was mapped to a co-folded region comprised of the N-terminal D-I of gH and gL, which is conserved in the MHV68 gH/gL model.

To analyze if the conserved fold of the MHV68 gH/gL complex is predictive of interactions with Eph family receptors, we performed pairwise co-precipitation analysis of MHV68 gH^ecto in the absence or presence of full-length MHV68 gL with each of the 14 full-length human Eph receptors expressed in 293T cells. MHV68 gH^ecto/gL interacted most efficiently with human (h) EphA4, with less pronounced binding to hEphB3 and hEphA6 (Fig 1B). We also observed weak signal for hEphA8, hEphA10 and hEphB1. However, this signal was consistent in the absence of MHV68 gL (Fig 1B), and observed with an Fc control (S1A Fig), indicative of non-specific binding. Among herpesviruses that interact with members of the Eph receptor family [26,27,29–34], KSHV exhibits the closest sequence similarity to both MHV68 gH and gL, based on sequence coverage and identity (Fig 1C). We therefore compared the interaction of hEphA4, a KSHV receptor and gH/gL interaction partner [29,32], with MHV68 and KSHV gH/gL. Precipitation of gH^ecto/gL from MHV68 or KSHV confirmed binding of both MHV68 gH/gL and KSHV gH/gL to hEphA4. On the other hand, MHV68 gH/gL did not interact with hEphA2, a KSHV-interacting Eph receptor (Fig 1D), in line with a previous report that did not observe an effect of hEphA2 on MHV68 fusion [55]. Eph receptors are an evolutionary conserved class of receptor tyrosine kinases and exhibit high amino acid identity between human and mouse homologs (Figs 1E, S1D). To analyze if this high inter-species conservation is reflected in observed MHV68 gH/gL-Eph interactions, we examined co-precipitation of MHV68 gH in the presence (Fig 1F) or absence (S1E Fig) of MHV68 gL with a subset of murine (m) Eph receptors. We observed a binding pattern recapitulating observations with human Eph receptors, identifying mEphA4 and mEphB3 as the most prominent interaction partners. This evolutionary conservation is likewise apparent in the interaction of KSHV gH/gL with mEphA2 and mEphA4 (S1F Fig). This is in line with mEphA2 receptor function shown for KSHV infection of murine pulmonary microvascular endothelial cells [31].

MHV68 genome annotations predict two coding sequences for MHV68 gL, starting from two alternative start codons. To determine potential differences in expression, gH complex formation and Eph interaction, we compared plasmids encoding the putative 173-amino-acid-protein (GenBank accession no. U97553) and 137-amino-acid-protein (containing a predicted 20 amino acid signal peptide) (GenBank accession no. AF105037) (S1C Fig). We did not observe differences in apparent molecular weights of expressed proteins, indicating cleavage after the signal peptide for generation of mature gL. In concordance, we did not see differences in gH/gL complex incorporation or EphA4 binding. These results agree with a previous report that identified expression of transcripts that align with AF105037 [22]. Subsequent gL amino acid positions are determined based on the 137-amino-acid-protein (S1C Fig).

To quantify binding of candidate Eph receptors to gH/gL and evaluate the requirement for additional cellular factors we analyzed the binding of purified murine Eph proteins to an immobilized MHV68 gL-gH fusion protein by enzyme-linked immunosorbent assay (ELISA) (Fig 1G). We determined an EC50 of 3.6 nM and 2.3 nM for mEphA4 and mEphB3, respectively, consistent with previously observed nanomolar binding affinities of KSHV gH/gL to EphA2 [55,56].

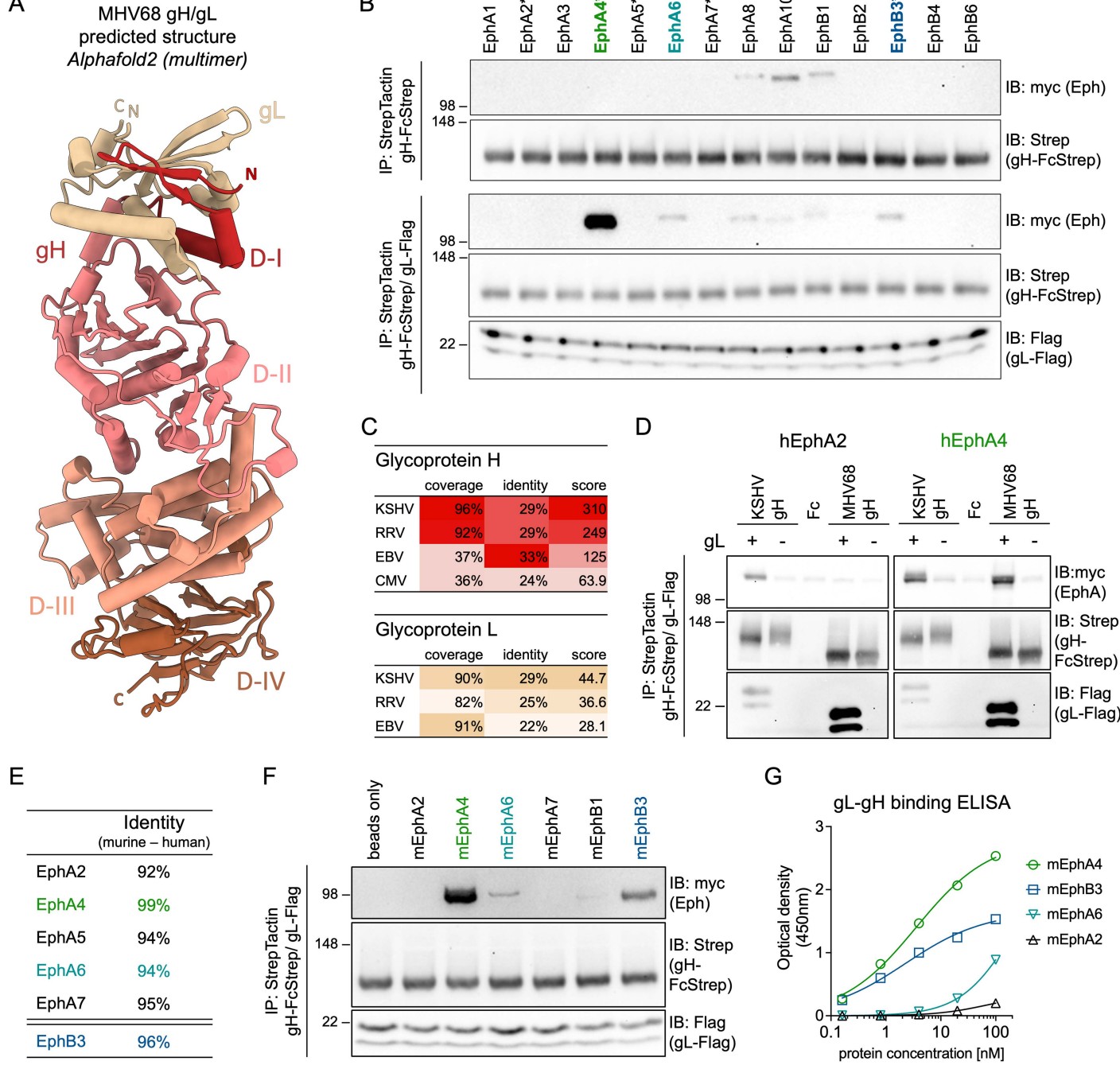

**Fig 1. The MHV68 gH/gL complex binds human and murine Eph receptors.** (A) The structure of the MHV68 gH/gL complex was predicted using alphafold2 multimer. MHV68 gH domains (D-I to D-IV) in the top ranked model were annotated based on EBV gH domain structure. Images were generated using ChimeraX. N and C termini of gH and gL are labeled "N" and "C", respectively. (B) Pairwise precipitation of soluble recombinant MHV68 gH ectodomain in complex with MHV68 gL (gH-FcStrep/gL-Flag) with individual human Eph proteins. MHV68 gH-FcStrep was used as control. Precipitates were analyzed by immunoblot with indicated antibodies. Asterisks indicate known KSHV, EBV or RRV gH/gL interaction partners. (C) Percentage of coverage, identical amino acids in aligned regions and total score of alignments of gH and gL sequences of Eph-interacting herpesviruses were determined by BLAST (Basic Local Alignment Search Tool). MHV68 gH or gL was used as query sequence. No alignment was returned for CMV gL. (E) Percentage of identical amino acids in aligned regions of human and murine Eph proteins as determined by BLAST. (F) Pairwise precipitation of soluble recombinant MHV68 gH ectodomain in

complex with MHV68 gL (gH-FcStrep/gL-Flag) with individual murine Eph proteins. Precipitates were analyzed by immunoblot with indicated antibodies. (G) Binding of dimeric, soluble, murine Eph proteins to an immobilized MHV68 gL-gH fusion protein was measured by enzyme-linked immunosorbent assay. Background corrected optical density at 450 nm is shown. Curve Fitting was performed using a four-parameter dose-response curve, determining an EC50 of 3.6 nM and 2.3 nM for mEphA4 and mEphB3 with a maximum OD of 2.8 and 1.6, respectively. For B, D, F, molecular weight is indicated in kDa.

These co-precipitations and ELISAs reveal a direct interaction of MHV68 gH/gL with multiple Eph receptors, with strongest binding to EphA4, followed by EphB3. EphA4 and EphB3 are receptors for KSHV and RRV, respectively, demonstrating an overlapping binding profile with both human and NHP GHV pathogens.

### Receptor function of MHV68 gH/gL-interacting Eph receptors

We next analyzed the biological and functional relevance of this interaction for the entry process and infection of MHV68. First, we employed a loss-of-function approach, using soluble murine Eph receptors as decoys to test their inhibitory potential on infection of permissive target cells with a GFP-expressing MHV68 reporter virus (MHV68 ORF59-GFP [62]) (Fig 2A). ORF59, an essential early gene, encodes the viral DNA polymerase processivity factor (vPPF), leading to expression of a vPPF::GFP fusion protein at early time points post *de-novo* infection [62,63]. NIH 3T3 murine fibroblasts, a highly susceptible cell type for MHV68 infection, exhibits broad Eph receptor expression, including *Epha4* and *Ephb3* (Fig 2B) in a published high throughput sequencing dataset [64]. To analyze the impact of competition with soluble Eph receptors, MHV68 ORF59-GFP inocula were incubated with a concentration series of soluble mEphA4, mEphA6 and mEphB3 prior to infection of NIH 3T3 cells. PBS, and soluble mEphA2 were used as controls (Fig 2C, 2D). In line with MHV68 gH/gL-Eph-interaction patterns observed in precipitation or ELISA, pre-incubation with soluble mEphA4 and mEphB3 reduced MHV68 infection of NIH 3T3 cells in a dose-dependent manner, while pre-treatment with mEphA2 and mEphA6 did not block infection (Fig 2C).

While both mEphA4 and mEphB3 inhibited MHV68 infection of NIH 3T3 cells, we observed differences between Eph receptors and murine (Fig 2D, 2E) and human cell types (Fig 2F, 2G). Pre-incubation of MHV68 virions with 100 nM homodimerized mEphA4 resulted in a near complete abrogation (~93%) of MHV68 infection on NIH 3T3 murine fibroblasts (Figs 2D, S2A), whereas infection of human fibroblasts (HFF) and endothelial cells (TIME) was inhibited by approximately 50–60% (Figs 2F, 2G, S2C, S2D). Soluble mEphA4 exhibited the lowest blocking efficiency on the murine endothelial cell line SVEC4–10 with an approximate 25% reduction of MHV68 infection (Figs 2E, S2B). Pre-incubation with soluble mEphB3 demonstrated a similar pattern to mEphA4, leading to an approximate 40% reduction of infection on murine NIH 3T3 fibroblasts (Fig 2C, 2D), while inhibition on human fibroblasts and endothelial cells only reached approximately 30% (Fig 2F, 2G). EphB3 pre-treatment had no effect on MHV68 infection of SVEC4–10 cells (Fig 2E). In concordance with interaction assays, we observed no inhibition of MHV68 infection after pre-incubation with mEphA2 on any of the tested cell lines.

In an orthogonal approach, we performed entry experiments using ectopic Eph overexpression in a non-permissive cell type (Fig 3A). Raji cells, an EBV-positive, human B lymphoblast cell line, support only low level MHV68 infection (<0.5% GFP+ cells at MOI 30, titrated on NIH 3T12), in agreement with low to absent mRNA expression of all 14 human Eph receptors (Fig 3B). To analyze contributions of single Eph receptors to MHV68 infection, Raji cells were transduced with lentiviruses encoding TwinStrep-tagged Eph receptors including human and murine EphA4 and EphB3 which bind MHV68 gH/gL (Fig 1) and inhibit infection on permissive cell lines (Fig 2). Empty vector, as well as human and murine EphA2 were used as negative controls. Immunoblot analysis of transduced and selected Raji cells confirmed expression of all analyzed Eph receptors, with lower levels of both human and mouse EphA4 observed compared to EphB3 and EphA2 (Fig 3C).

Ectopic expression of EphB3 of human and mouse origin exhibited the strongest enhancement of MHV68 ORF59-GFP infection, leading to an ~20-fold increase in GFP+ cells. Expression of EphA4 had a comparably less pronounced effect on

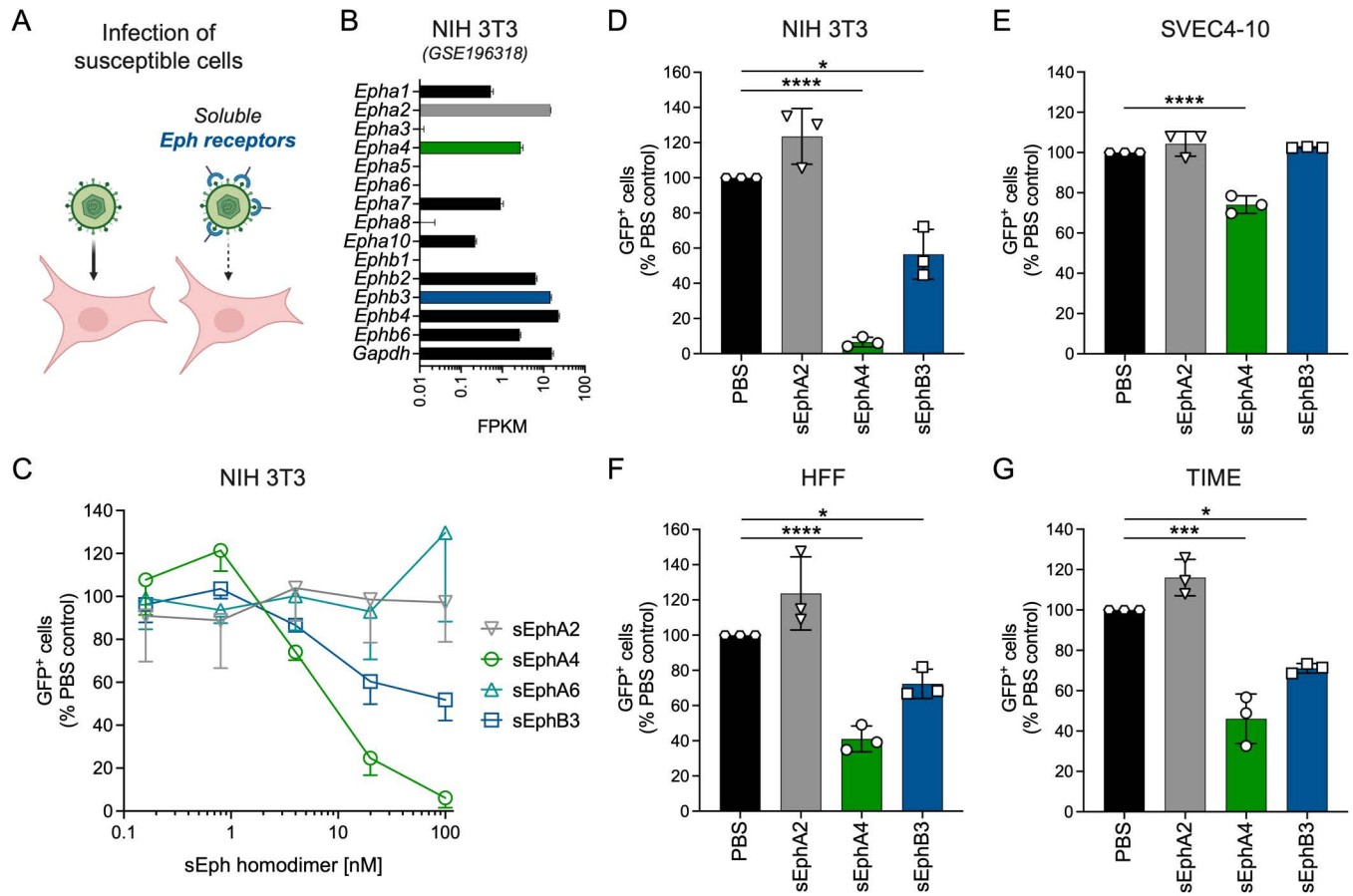

**Fig 2. Soluble murine EphA4 and EphB3 inhibit MHV68 infection of endothelial cells and fibroblasts.** (A) Schematic of Eph receptor-dependent block of MHV68 infection of susceptible cell lines (BioRender.com/jdlijfo). (B) Normalized read counts of the 14 *Eph* receptor genes in NIH 3T3 cells (GEO dataset series GSE196318 [64]). (C) Dose-dependent inhibition of MHV68 infection by soluble murine EphA4-Fc and EphB3-Fc but not EphA6-Fc on NIH 3T3 murine fibroblasts. MHV68 ORF59-GFP was pre-incubated with murine EphA4-Fc, EphA6-Fc or EphB3-Fc. EphA2-Fc and PBS were used as controls. GFP expression was measured by flow cytometry. Infection is indicated as percentage of GFP+ cells normalized to PBS controls. Mean of three independent experiments, error bars represent SD. (D-G) Cell type-dependent inhibition of MHV68 infection by soluble murine Eph proteins at 100 nM homodimerized protein. EphA2-Fc and PBS were used as controls. GFP expression as indicator of infection was measured by flow cytometry. Infection is shown as percentage of GFP+ cells normalized to PBS controls. Mean and symbols represent three individual experiments, error bars represent SD. Statistical significance was evaluated by ordinary one-way ANOVA, corrected by Holm-Šídák's multiple comparisons test. *: p-value < 0.05, ***: p-value < 0.001, ****: p-value < 0.0001.

MHV68 infection of ~8-fold enhancement compared to empty vector. This might be reflective of consistently lower expression levels of EphA4 constructs compared to EphB3. Nevertheless, overexpression of EphA4 led to an equivalent species-independent increase in MHV68 ORF59-GFP infection (Fig 3D, 3E). In summary, orthogonal soluble blockade and single receptor entry assays demonstrate that EphA4 and EphB3 act as functional entry receptors for MHV68.

### The MHV68 gH/gL-Eph interaction is mediated by a conserved gammaherpesviral Eph-interaction motif

The gH/gL-Eph interaction is dependent on a structural motif formed by D-I of gH and gL that contacts residues of Eph receptors that mediate binding of the GH loop in ephrin ligands [55,56,65,66]. Structure predictions of the MHV68 gH/gL heterodimer in complex with either the murine EphA4 or EphB3 ligand binding domain (LBD) were performed using alphafold2 multimer, refined using Rosetta and aligned with the previously published structure of the KSHV gH/gL-EphA2

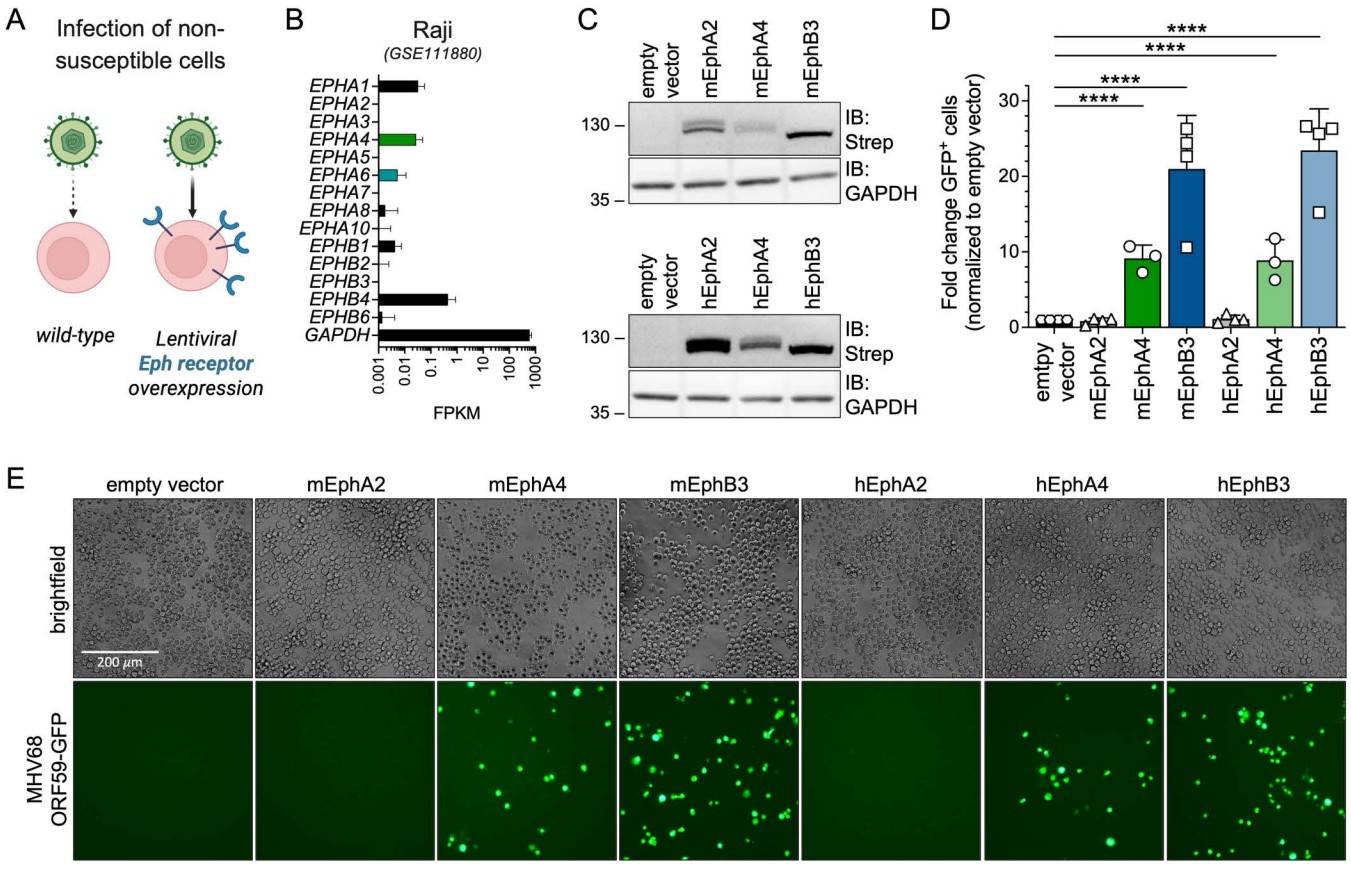

**Fig 3. Overexpression of human and murine Eph receptors enables MHV68 infection of non-permissive Raji B cells.** (A) Schematic of Eph overexpression-dependent enhancement of MHV68 infection of non-susceptible cell lines (BioRender.com/cuc5pc0). (B) Normalized read counts of the 14 *Eph* receptor genes in Raji B cells (GEO dataset series GSE111880 [90]). (C) Raji cells were transduced with TwinStrep-tagged murine (m) or human (h) Eph expression constructs or an empty vector (eV) control and selected by antibiotic resistance. Lysates of transduced Raji cell pools were analyzed for Eph-Strep expression by immunoblot. Molecular weight is indicated in kDa. (D) Transduced Raji cells analyzed in (C) were infected with MHV68 ORF59-GFP and GFP expression was measured by flow cytometry (primary infection in empty vector controls: 0.22–0.49% GFP+ cells). Infection is indicated as fold change of GFP+ cells normalized to empty vector controls. Mean and symbols represent three-four individual experiments, error bars represent SD. Statistical significance was evaluated by ordinary one-way ANOVA, corrected for multiple comparisons by Holm–Šídák's multiple comparisons test. ****: p-value < 0.0001. (E) Micrographs show representative infection of the indicated cell pools quantified in (D).

complex (PBDid 7B7N) (Fig 4A–4C). MHV68 gH/gL exhibited an Eph binding interface reminiscent of EBV and KSHV gH/gL-EphA2 complexes, with contacts observed throughout the N-terminal region of gL. As gL is co-folded with gH D-I and separated from D-II by a linker helix we sought to analyze if a complex consisting of gH D-I and full-length gL is sufficient to mediate the gH/gL-Eph interaction. We performed co-precipitation analyses of the constructs encoding the MHV68 gH ectodomain (gH$^{ecto}$) or MHV68 gH D-I and the D-I/D-II linker sequence (gH$^{D-I}$) in the absence or presence of full-length MHV68 gL with murine EphA4. MHV68 gH$^{D-I}$ efficiently incorporated gL to higher levels than gH$^{ecto}$ and bound EphA4 (Fig 4D). The predicted conservation of intermolecular hydrogen bonds between MHV68 gH and EphA4/ EphB3 indicated the aspartic acid at position 52 (D52$^{MHV68\ gH}$) as a key interacting residue contacting arginine 106 on both mEphA4 and mEphB3 (R106$^{EphA4/\ EphB3}$) (Fig 4B, 4C). R103$^{EphA2}$, the conserved arginine in the ephrin binding channel of EphA2, has been reported as single contact site for KSHV gH, forming a salt bridge with glutamic acid 52 (E52$^{KSHV\ gH}$) [56], the positional equivalent of D52$^{MHV68\ gH}$ (S3A Fig). Analysis of hydrogen bonds formed between MHV68 gH and gL, identified Y50$^{MHV68\ gH}$ as a potential stabilizing residue for the gH-gL interaction and K24$^{MHV68\ gL}$ as a potential contact residue for Eph receptors (Fig 4B, 4C). We

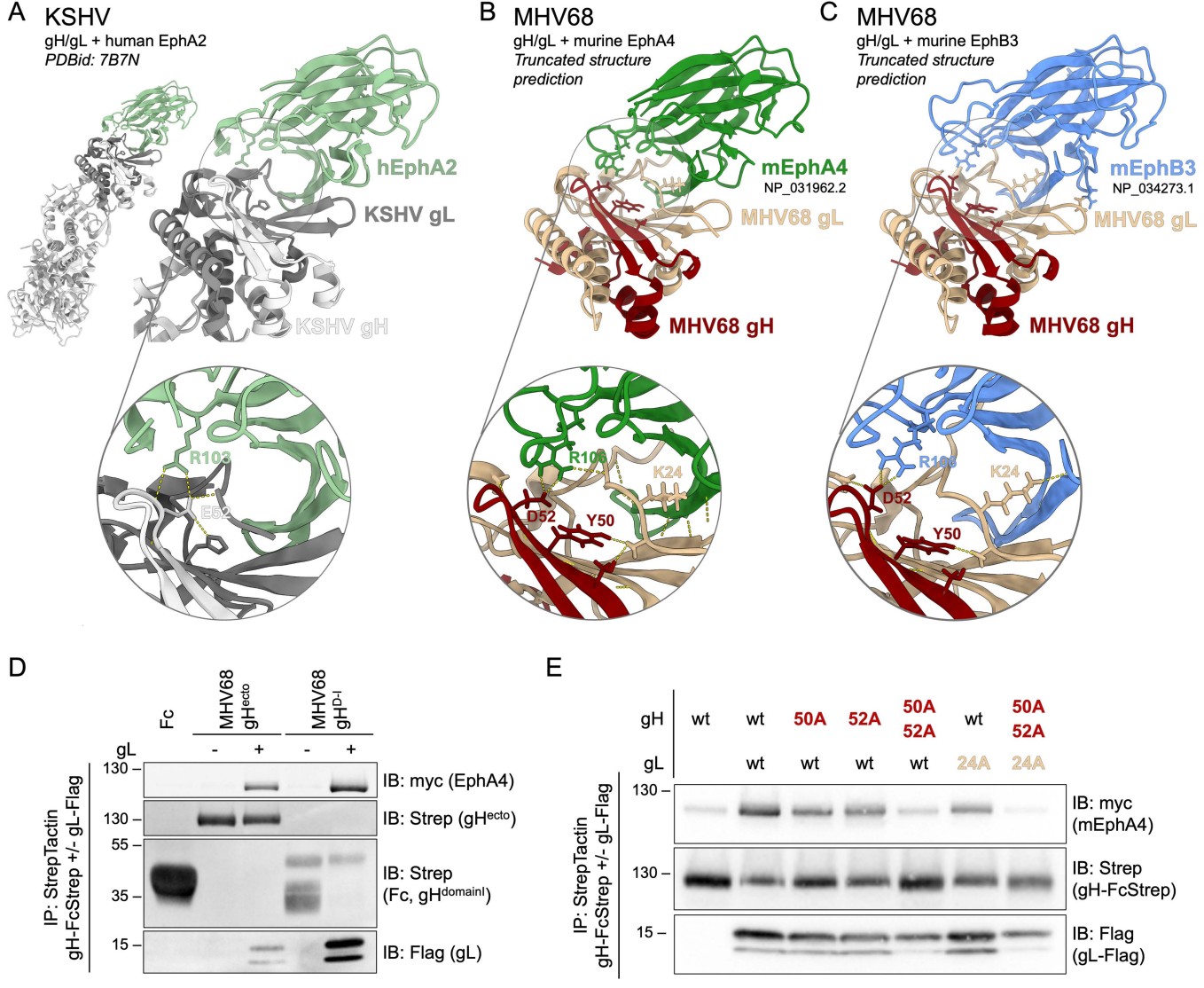

**Fig 4. MHV68 gH/gL shares a structurally conserved Eph-interaction motif with the KSHV gH/gL complex.** (A) Reference structure of KSHV gH/gL in complex with human EphA2 (PDBid: 7B7N). Images were generated using ChimeraX. The core interacting residue pair E52$^{KSHV\ gH}$ - R103$^{EphA2}$ is depicted in inset. (B-C) Truncated MHV68 gH (NP_044860.1) and MHV68 gL (NP_044884.3) were predicted in complex with murine EphA4 (NP_031962.2) (B) or murine EphB3 (NP_034273.1) (C) using alphafold2 (multimer, n = 15). Resulting models were aligned against the known structure of KSHV gH/gL in complex with human EphA2 (PDBid: 7B7N). Images were generated using ChimeraX. Putative residues involved in Eph interaction or complex stabilization (Y50$^{MHV68\ gH}$, D52$^{MHV68\ gH}$, K24$^{MHV68\ gL}$) and R106$^{EphA4/EphB3}$ are depicted in insets. Dashed lines represent predicted hydrogen bonds. (D) Pairwise precipitation of soluble recombinant MHV68 gH ectodomain or MHV68 gH D-I in complex with MHV68 gL (gH-FcStrep/gL-Flag), with murine EphA4. Fc and MHV68 gH constructs without gL were used as control. Precipitates were analyzed by immunoblot with indicated antibodies. (E) Putative interaction residues Y50$^{MHV68\ gH}$, D52$^{MHV68\ gH}$, and K24$^{MHV68\ gL}$ were mutated to alanine in single point mutants or a gH double point mutant. Combinations of MHV68 gH ectodomain and gL point mutants were precipitated with murine EphA4. MHV68 gH-FcStrep was used as control. Precipitates were analyzed by immunoblot with indicated antibodies. (D-E) Molecular weight is indicated in kDa.

next evaluated the impact of these residues on gH/gL-Eph complex formation. Single amino acids in the putative interaction interface were mutated to alanine and subjected to co-precipitation of wt and mutant combinations with murine EphA4 (Figs 4E, S3). Single point mutations of predicted interaction residues Y50$^{MHV68\ gH}$, D52$^{MHV68\ gH}$ and K24$^{MHV68\ gL}$ led to comparable

levels of gH expression and gL incorporation. In contrast to KSHV, in which a single point mutation in E52$^{MHV68\ gH}$ leads to complete abrogation of EphA2 interaction, single point mutations in MHV gH and gL only led to a moderate reduction in EphA4 binding in our assays. However, the combination of mutations Y50A$^{MHV68\ gH}$, E52A$^{MHV68\ gH}$ and K24A$^{MHV68\ gL}$ reduced EphA4 binding to background levels (Fig 4E). The double point mutation in Y50-D52$^{MHV68\ gH}$ led to decreased incorporation of the low molecular weight gL. Taken together our data indicates that the MHV68 gH/gL-Eph interaction is shaped by the interaction of D52$^{MHV68\ gH}$ and R106$^{EphA4/\ EphB3}$ and a larger gL-Eph interaction interface.

## MHV68 infected mice generate neutralizing antibodies targeting D-I of gH

The gH/gL complex has been identified as a major target for GHV neutralizing antibodies *in vivo* [19–23]. To analyze the effect of depletion of gH/gL-directed antibodies on serum neutralization capacity, we inoculated C57BL/6 mice with 1000 PFU wild-type (WT) MHV68 by the restrictive intranasal route of infection that models oropharyngeal GHV infection of people (Fig 5A). At 28 dpi, serum was collected from two independent groups of WT infected mice and analyzed by ELISA using plates coated with purified gL-gH fusion protein. WT infection elicited robust gH/gL-targeting antibody responses (Fig 5B). We next evaluated the serum neutralizing capacity on MHV68 infection of NIH 3T3 fibroblasts by analyzing reduction in GFP$^+$ cells after pre-incubation of MHV68 ORF59-GFP with pooled, heat-inactivated sera (Fig 5C). We observed similar neutralizing capacity of serum from both MHV68 WT infected groups when compared to pre-incubation with serum from naïve mice. To evaluate the relative contribution of gH/gL-targeting nAbs to MHV68 neutralization, we depleted gH/gL-specific antibodies from serum of naïve or MHV68 WT infected mice using three rounds of depletion with either gH$^{ecto}$/gL or gH$^{D–I}$/gL complexes pre-coupled to magnetic beads. Adsorption using gH$^{ecto}$/gL or gH$^{D–I}$/gL decreased gH/gL-specific antibodies as measured by ELISA, but we observed slightly higher residual antibody levels in gH$^{D–I}$/gL-depleted samples (S4A Fig). To measure the neutralizing capacity of sera following adsorption, serum samples were pre-incubated with MHV68 ORF59-GFP prior to infection of NIH 3T3 cells. Depletion of gH/gL-targeting antibodies reduced neutralizing activity by approx. 50% with no pronounced differences between gH$^{ecto}$/gL or gH$^{D-I}$/gL (Figs 5D, 5E, S4B). These findings indicate that the gH/gL interface is a primary target for gH/gL nAbs.

## Discussion

In this study, we demonstrate that Eph receptors are functional entry receptors for MHV68, indicating strong evolutionary conservation with other human and primate herpesviruses. We identify EphA4 and EphB3 as direct interaction partners of the MHV68 gH/gL complex that facilitated MHV68 infection in different cell types from humans and mice. Mice infected with MHV68 mounted neutralizing antibody responses that target the receptor binding domain comprised of gH D-I and gL.

EphA4 and EphB3 are known GHV entry receptors. The gH/gL complexes of both KSHV and RRV bind EphA4 [29,32,33]. EphB3, has been identified as high affinity receptor for RRV, a rhadinovirus of rhesus macaques but so far, no interaction with human herpesviruses has been observed [33]. The role of EphA4 in KSHV infection seems to be complex since EphA4 expression enhances KSHV infection and gH/gL-dependent cell fusion [29], but knock-down in permissive SLK cells similarly increased KSHV infection [32]. Unlike a previous study that reported a modest enhancement of MHV68 fusion by murine EphA2 [55], neither human nor murine EphA2 bound to MHV68 gH/gL or influenced MHV68 infection in our assays.

While the Eph-binding profiles of herpesviruses do not exhibit a complete overlap, the mechanism of binding via a binding domain comprised of gH D-I and gL that contacts the ephrin ligand binding pocket on Eph receptors appears conserved across gH/gL complexes [55,56,66]. Our data demonstrates that D-I of MHV68 gH in complex with gL is likewise sufficient to mediate the gH/gL interaction. The interaction between R103$^{EphA2}$ and E52$^{KSHV\ gH}$, which is essential for the nanomolar affinity binding of KSHV to EphA2, is conserved in structural models of MHV68 gH/gL interactions with EphA4 and EphB3. The similarities between the structural predictions of MHV68 gH/gL-EphA4 and experimental models of KSHV gH/gL-EphA2 are consistent with the nanomolar binding affinities observed for MHV68 gH/gL interactions with mEphA4 and mEphB3 in ELISA.

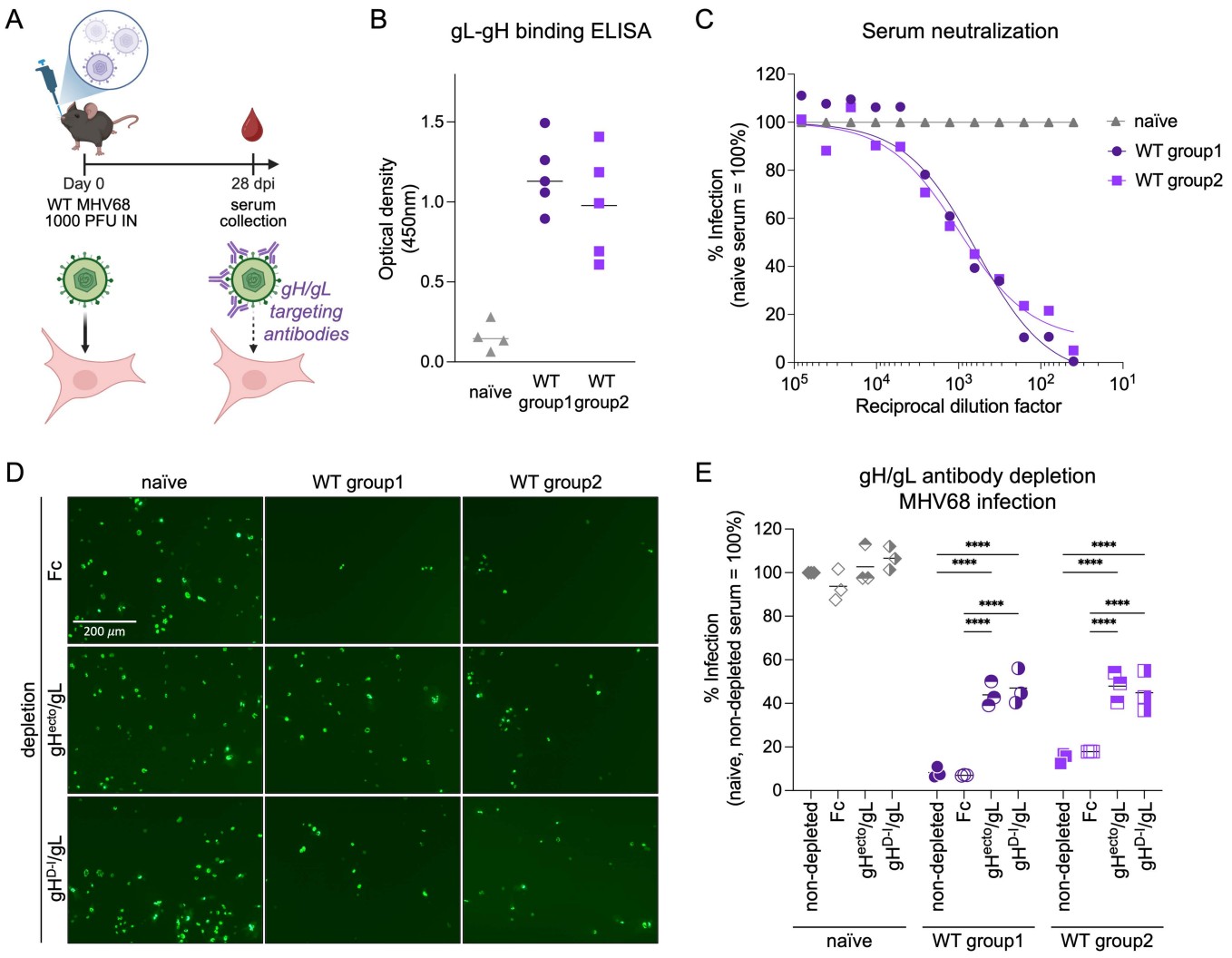

**Fig 5. Neutralizing antibodies in MHV68 infected mice target the MHV68 gH/gL complex.** (A) Schematic of virus neutralization by gH/gL-targeting nAbs. C57BL/6 mice were infected with 1,000 PFU MHV68 WT by intranasal inoculation. Serum was collected at 28 dpi (BioRender.com/6j9fwj6). (B) gH/gL-specific IgG from naïve or MHV68-infected C57BL/6 at a serum dilution of 1:80 was measured by MHV68 gL-gH ELISA. Background corrected optical density at 450 nm is shown. (C) Serum of MHV68-infected C57BL/6 mice neutralizes MHV68 infection on NIH 3T3 cells. MHV68 ORF59-GFP was pre-incubated with mouse serum at the indicated dilution. GFP expression was measured by flow cytometry. Infection is indicated as percentage of GFP+ cells normalized to naïve serum. (D-E) Serum neutralization of MHV68 infection on NIH 3T3 cells is mediated by gH/gL-targeting antibodies. Antibodies to gH$^{ecto}$/gL or gH$^{D-I}$/gL were depleted using soluble complexes pre-coupled to magnetic beads. Fc was used as control. Neutralization was analyzed as in (C) at a serum dilution of 1:80. Micrographs were taken at 16 hpi. Mean and symbols representing individual experiments are shown. Statistical significance was evaluated by ordinary two-way ANOVA corrected for multiple comparisons by Tukey's multiple comparisons test. ***: p-value < 0.001, ****: p-value < 0.0001.

Structural and binding analyses identify D52$^{MHV68\ gH}$ as a key point of contact with Eph receptors even though co-precipitation analysis indicated a higher residual binding of single D52$^{MHV68\ gH}$ point mutants than observed in studies using E52$^{KSHV\ gH}$ point mutants. Mutation of Y50$^{MHV68\ gH}$ and K24$^{MHV68\ gL}$ similarly resulted in a reduced EphA4 interaction. While K24$^{MHV68\ gL}$ was predicted to directly interact with Eph receptors in our structural models, the contact of Y50$^{MHV68\ gH}$ with MHV68 gL potentially stabilizes the N-terminal region of gL that is predicted to form a large Eph interaction interface. Combination of all three identified interaction sites abrogated EphA4 interaction but led to a reduced incorporation of gL.

This reduced gL incorporation might be indicative of a destabilization of the gH/gL complex and could in turn affect EphA4 interaction. Alternatively, Eph interaction could lead to a secondary stabilization of the gH/gL complex, with loss of Eph binding leading to decreased gH/gL stability. In summary, our data supports a conserved role of the structurally conserved D52[MHV68 gH] as a key contact with Eph receptors that is supported by additional gL-Eph contact sites as described in KSHV and EBV [55,56]. Future investigations will determine how differences in binding interfaces and affinity affect Eph receptor targeting and GHV infection.

Disruption of the Eph-gH/gL association impacts GHV infection and tropism in cell culture [26,27,30–33,67]. Our data demonstrates that soluble mEphA4 and mEphB3 inhibit MHV68 infection in both endothelial cells and fibroblasts, with mEphA4 generally showing greater blocking efficiency. We observed a slight but consistent increase in infection upon sEphA2 treatment, which did not reach significance. In the future, a more detailed analysis of Eph and ephrin expression in different target cell types used, along with complementary blocking experiments, may help clarify the technical or biological relevance of this finding. Observed inhibitory effects did not exhibit a clear cell type-dependent pattern. While human HFF fibroblasts and TIME endothelial cells exhibited a comparable Eph-dependency, murine NIH 3T3 fibroblasts and SVEC4–10 endothelial cells exhibited the most and least pronounced responses, respectively. These differences are reminiscent of previous studies that pointed to variable receptor usage for different B cell lines in cell culture [30,68,69] highlighting the challenges of studying virus-host interactions in immortalized cell lines or donor-specific primary cells and support the need for *in vivo* analyses. The role of Eph-gH/gL interactions in cell culture systems [26,27,29–32] only partially recapitulate *in vivo* observations. For example, RRV and MHV68 gL-negative mutant viruses exhibited strong entry defects in cell culture systems but were able to establish latency *in vivo* at levels comparable to WT virus [52,70]. Nevertheless, gL is conserved across GHV and certainly provides a competitive advantage *in vivo*. MHV68 gL influences infection and latency establishment in a route-dependent manner [54]. As receptor interactions are known to shape GHV tropism, these different roles might stem from the multiple target cell types encountered as the virus transverses restrictive mucosal barriers and disseminates to sites of latency in secondary lymphoid tissues [48,71]. MHV68 infection of mice thus offers a genetically tractable model to assess the *in vivo* relevance of Eph receptors in primary infection via different routes of dissemination and latency establishment. This study lays the groundwork for combinatorial approaches using viral mutants disrupted in the Eph interaction motif or gL incorporation, and lineage-specific knock-outs of Eph receptors.

The Eph-ephrin system exhibits a complex pattern of spatial and temporal expression *in vivo*, which is functionally refined by the bidirectional signaling induced upon receptor-ligand interactions. Receptor activation leads to recruitment of homotypic as well as heterotypic Eph receptors to form higher order signaling clusters [72]. Accordingly, overall Eph expression profiles might more accurately predict functional outcomes of receptor engagement than expression of any given singular receptor. To reduce the complexity of the Eph/ephrin system and analyze the contribution of individual Eph receptors to MHV68 infection we employed ectopic overexpression of Eph receptors on Raji B cells, that possess low endogenous Eph levels. Interestingly, differences in apparent saturation binding of EphA4 and EphB3 observed in co-precipitation and ELISA experiments did not correlate with augmentation of MHV68 infection of Raji cells. Given the slightly higher EC50 alongside a higher maximum OD for mEphA4 binding compared to mEphB3 in gL-gH ELISAs, this might be reflective of differences in binding affinity and complex stability. While EC50 reflects apparent binding affinity, maximum OD represents saturation binding, suggesting that mEphB3 may exhibit higher initial affinity, whereas mEphA4 achieves greater total binding at steady state. Alternatively, these differences could be attributed to consistently lower expression levels of EphA4 when compared to EphB3 or EphA2, potentially hinting at an intrinsic difference in obtainable EphA4 levels in Raji B cells. Eph/ephrin crosstalk has been implicated in the maintenance of the hemopoietic stem cell (HSC) niche, with higher Eph expression levels in undifferentiated cells. Of note, EphA4 does not follow this pattern since transcripts were detected in human B cell progenitor cells as well as mature B cells [73,74]. Nevertheless, established B cell lines exhibit low EphA4 expression compared to other cell types, such as endothelial cells and fibroblasts. Interestingly, EBV LMP1 is a known regulator of EphA4 expression, and an inverse correlation of EBV infection and EphA4

expression was observed in EBV+ and EBV− diffuse large B-cell lymphoma (DLBCL) and EBV− tonsils compared to EBV+ posttransplant lymphoproliferative disorder. Functionally, low EphA4 levels were linked to sustained proliferation of lymphoblastoid cell lines and poor prognosis in DLBCL patients [75]. Similarly, ring finger protein 5 (RNF5), a pro-oncogenic factor in multiple malignancies [76–78] has been reported to downregulate EphA3 and EphA4 expression in PEL cell lines associated with higher ERK and Akt activation [79]. Eph receptor engagement by ephrin ligands is often associated with receptor internalization and subsequent proteosomal degradation [80]. It remains to be determined if gH/gL-dependent MHV68 infection similarly triggers a decrease in EphA4 expression, in addition to possible effects on the proliferation and survival of MHV68 infected B cells.

The gH/gL complex has been identified as a key target for neutralizing humoral responses in EBV+ or KSHV+ patients [19–21,81] and animal models [22,23]. Our study corroborates the efficient generation of gH/gL-directed antibodies upon MHV68 WT infection, which contribute to in vitro MHV68 neutralization capacity. These results are in line with previous studies that mapped the gH/gL-directed neutralizing capacity to the full gH/gL complex but not gH alone [22]. We are the first to show that adsorption with gH/gL complexes encompassing the complete ectodomain or restricted to gH D-I led to comparable loss in MHV68 neutralization. As D-I mediates the gH/gL-Eph receptor interaction, a disruption of Eph receptor interactions as mechanistic target of gH/gL-directed nAbs is likely. Similarly, analyses of monoclonal nAbs that target the EBV gH/gL complex point to gH D-I as one site of vulnerability targeted by multiple human antibodies isolated from healthy, EBV+ donors [81]. Curiously, we did not observe a complete depletion of gH/gL-directed antibodies from sera of infected mice, reminiscent of a recent study that demonstrated complete absorption from a patient in KS remission, with residual anti-gH/gL antibodies in the serum of an asymptomatic patient [20]. Incomplete antibody depletion may reflect limitations in antigen presentation on beads or the presence of low-affinity antibodies. Further investigation is needed to determine whether these antibodies recognize distinct epitopes or might constitute low affinity binders unable to engage under depletion conditions. In addition, the lack of complete neutralization by gH/gL-directed antibodies may be attributed to other glycoprotein/ receptor interactions yet to be defined.

While aspects of KSHV/ EBV entry in cell culture systems and GHV infection in vivo have been analyzed independently, highly analogous animal model pathogens provide powerful systems to analyze virus-receptor interactions, the role of entry receptor in cell tropism, and the impact of targeted disruption of these interactions on the establishment and maintenance of chronic, life-long infection in vivo. Our study is the first to characterize a conserved function of Eph receptors in MHV68 infection. The biological significance of this interaction was demonstrated by the key finding that neutralizing antibodies which target the Eph-interaction domain inhibit MHV68 infection. This further validates and expands the utility of MHV68 infection of mice as a model system for GHV infection and vaccine development. Future studies on the protection offered by MHV68 gH/gL-targeting antibodies in vivo and rationally designed vaccines targeting receptor-virus interaction domains will inform optimized vaccination strategies against human pathogenic GHVs.

## Materials and methods

### Ethics statement

All animal procedures reported in this study that were performed by NCI-CCR affiliated staff were approved by the NCI Animal Care and Use Committee (ACUC) and in accordance with federal regulatory requirements and standards (internal protocol number HAMB-002, approved 10/03/2022). All components of the intramural NIH ACUC program are accredited by AAALAC International.

### Animal studies

Female C57BL/6 mice were purchased from Charles River Laboratories (Wilmington, MA). Seven-week-old mice were anesthetized using 1–4% isoflurane and inoculated intranasally (IN) with 1,000 PFU WT MHV68 diluted in 20 µl Dulbecco's Modified Eagle Medium (DMEM), high glucose (Thermo Fisher Scientific, Waltham, MA; Corning, Corning, NY)

supplemented with 10% FBS (R&D Systems, Minneapolis, MN), 100 U/mL penicillin, 100 µg/mL streptomycin (Corning), and 2 mM L-glutamine (Corning) (D10). On day 28 post-infection, blood was collected following humane euthanasia using isoflurane by post-mortem cardiac puncture, transferred to serum separator tubes (BD, Franklin Lakes, NJ) and centrifuged (21,100 × g, 20 min). Serum was carefully collected, aliquoted and stored at -80°C.

## Cell culture

Human embryonic kidney (HEK) 293T cells (RRID:CVCL_0063), human foreskin fibroblasts (HFF) (RRID:CVCL_XB54) and SVEC4–10 (RRID:CVCL_4393) murine endothelial cells were cultured in D10. NIH 3T3 (RRID:CVCL_0594) and NIH 3T12 (ATCC CCL-164, Manassas, VA) murine fibroblasts were maintained in DMEM supplemented with 8% FCS, 100 IU/mL penicillin, 100 µg/mL streptomycin, and 2 mM L-glutamine (D8). TIME (RRID:CVCL_0047) human telomerase-immortalized human microvascular endothelial cells were cultured in Vascular Cell Basal Medium (ATCC PCS-100–030) supplemented with Microvascular Endothelial Cell Growth Kit-VEGF (ATCC PCS-110–041) and blasticidin (Invivogen, San Diego, CA) at 12.5 µg/ml. Raji cells (RRID:CVCL_0511) were cultured in RPMI (Thermo Fisher Scientific, Corning) supplemented with 10% FCS, 100 U/mL penicillin, 100 µg/mL streptomycin, and 2 mM L-glutamine (R10). All cells were cultured at 37°C in 5% $CO_2$.

## Plasmids and recombinant proteins

pcDNA4 vectors expressing full-length human Eph constructs were previously described [33]. pCMV3-C-Myc vectors containing full-length constructs for murine EphA2, EphA4, EphA6, EphA7, EphB1 and EphB3 were purchased from Sino Biological (S1 Fig for accession numbers). Sequence similarity with orthologs was determined using the BLAST algorithm [82]. The expression construct for the soluble, ectodomain of MHV68 gH (gH^ecto), endcoded by *orf22* fused to the Fc fragment of human IgG (gH-FcStrep) contains the codon-optimized sequence (see S1 Table) coding for the predicted extracellular domain of MHV68 gH (amino acids 25–702, as predicted by DeepTMHMM) (analogously to [30,83]) fused to the carboxy terminus of the 21-amino-acid murine immunoglobulin G kappa subunit ([IgG(κ)] signal peptide (PIR locus KVMS32) and the amino terminus of the Fc part from human IgG1 (GenBank accession no. S72664, amino acids 146–374) followed by a Twin-Strep-tag and 6xHis-tag. MHV68 gH ectodomain single point mutants and a truncated construct encoding D-I and the D-I/D-II linker sequence (gH^D-I, amino acids 25–106) were generated using site-directed mutagenesis. The expression construct for MHV68 gL, encoded by *orf47* contains the codon-optimized sequence coding for full-length MHV68 gL (NP_044884.3, see S1 Table) followed by a single C-terminal Flag tag. An N-terminally truncated MHV68 gL expression construct lacking the first 36 amino acids (gL^ΔN36) and single point mutants were generated using site-directed mutagenesis. Mutations were verified by Sanger Sequencing of the targeted region. See S1 Table for a complete list of primers.

Recombinant soluble Eph receptors were purchased as Fc fusion proteins from R&D Systems (#639-A2, #641-A4, #607-A6, #432-B3). Cloning, protein expression and purification of the MHV68 gL-gH fusion protein was performed at the CCR Protein Expression Laboratory (PEL). Synthesized DNA, gene-optimized for mammalian expression, was used to create Entry clones containing the DNA sequences for gL-Linker-gH-Linker-His6. An Entry clone containing the CMV51 promoter sequence was created by PCR using pDest-720 (Sabine Geisse, Novartis) as template DNA and pDonr-233 (Esposito lab) as backbone vector via Gateway recombination [84]. Final Expression clones were created by assembling promoter and gene sequence Entry clones by Gateway multisite reactions as described before [85] using pDest-303 (Addgene #159678) as destination vector. Expi293F (Thermo Fisher Scientific) culture supernatant was harvested 96 h after transfection and stored at -80°C prior to protein purification. The culture supernatant was thawed, dialyzed against 20 mM HEPES, pH 7.4, 300 mM NaCl (Buffer A) and filtered through a bottle top 0.45 µm filter. Imidazole was added to a final concentration of 25 mM. All purification steps were performed on a Bio-Rad NGC chromatography system at room temperature. A 2 x 1-ml HisTrap FF column (Cytiva, Marlborough, MA) was equilibrated in 10 column volumes (CV) 98%

Buffer A and 2% Buffer B (20 mM HEPES, pH 7.4, 300 mM NaCl, 500 mM imidazole). Filtered culture supernatant was loaded onto the column followed by 10 CV wash in 2% Buffer B. Protein was eluted from the column with bump to 100% Buffer B. Fractions were collected across the gradient and analyzed by SDS-PAGE/Coomassie Blue staining. Elution fractions containing target protein were pooled.

### Virus stock production and titration

Viruses utilized in this study include WT MHV68 (WUMS strain) for infection of C57BL/6 mice and bacterial artificial chromosome (BAC)-derived MHV68 ORF59-GFP. The MHV68 DsRed bacmid carrying a C-terminal GFP fusion to vPPF (ORF59) was previously described [62]. Virus passage and titer determination on NIH 3T12 fibroblasts were performed as previously described [86]. In short, 3T12 cells were infected at low MOI conditions (MOI 0.01 – 0.05) for virus stock production. Cells were harvested 8 dpi and homogenized using dounce homogenizers on ice. Supernatant was clarified by two sequential centrifugation steps at 1,600 rpm for 7 min and 4,000 x g for 15 min. Virus was concentrated from clarified supernatant by high-speed centrifugation (14,000 x g, 2 h) and resuspended in 1:80 of original cell culture volume. Aliquots were stored at -80°C. Titers were determined by quantification of plaque formation on NIH 3T12 cells. NIH 3T12 cells were plated at a density of $1 \times 10^5$ cells per ml. One day after plating, cells were incubated with serial dilutions of viral inocula for 1 hr at 37°C, with rocking every 15 min, followed by an overlay of 1.5% (w/v) methylcellulose (Sigma-Aldrich, Saint Louis, MO) in DMEM supplemented with 5% FCS, 100 IU/mL penicillin, 100 µg/mL streptomycin. After 7–8 d incubation at 37°C, cells were fixed with 100% methanol (Sigma- Aldrich), stained with 0.1% crystal violet (Sigma- Aldrich) in 20% methanol, and plaques were scored.

### Lentivirus production and transduction

For production of lentiviral particles, 10 cm cell culture grade petri dishes of approximately 80% confluent 293T cells were transfected with 3 µg each of pLP1 (HIV-1 gag and pol), pLP2 (HIV-1 rev), pLP/VSVG (VSV G glycoprotein (VSV-G)) and lentiviral expression constructs (pLenti CMV Blast DEST (706–1), pLenti-CMV-Blast-hEphA2-Strep, pLenti-CMV-Blast-hEphA4-Strep, pLenti-CMV-Blast-hEphB3-Strep, pLenti-CMV-Blast-mEphA2-Strep, pLenti-CMV-Blast-mEphA4-Strep, pLenti-CMV-Blast-mEphB3-Strep) using Transporter 5 Transfection Reagent (Polysciences, Warrington, PA) as per manufacturer instructions. The supernatant containing the pseudo-typed lentiviral particles was harvested 2–3 d after transfection and filtered through 0.45 µm membranes (Merck Millipore, Burlington, MA). For transduction, lentivirus stocks were added at a 1:5 dilution in the presence of 8 µg/ml polybrene, followed by centrifugation (1,500 x g, 1 h, RT). After 48 h, the selection antibiotic blasticidin (Invivogen) was added to a final concentration of 10 µg/ml.

### Immunoprecipitation and immunoblot analysis

For interaction analysis of gH/gL complexes with Eph receptors, 293T cells were transfected using Transporter 5 Transfection Reagent (Polysciences), per manufacturer instructions. Lysates of 293T cells transfected with expression constructs for human or mouse Eph receptors were prepared 2 days after transfections in NP40 lysis buffer (1% Nonidet P40 Substitute (Sigma-Aldrich), 150 mM NaCl (Quality Biological, Gaithersburg, MD), 50 mM HEPES (Quality Biological), 1 mM EDTA (Crystalgen) with freshly added cOmplete Mini Protease Inhibitor Cocktail (Roche, Indianapolis, IN) and PhosSTOP Phosphatase Inhibitor Cocktail (Roche). For input immunoblot, 2 µl of each lysate was denatured in 1x NuPAGE LDS sample buffer (Thermo Fisher Scientific) at 70°C for 10 min, separated by polyacrylamide gel electrophoresis (PAGE) using 8–16% Tris-Glycine polyacrylamide gradient gels (Thermo Fisher Scientific) with Tris-Glycine SDS running buffer (25 mM Tris, 192 mM glycine, 0.1% SDS) and transferred to Polyvinylidendifluorid (PVDF) membranes (iBlot2). The membranes were blocked in 1x Fluorescent blocking buffer (Thermo Fisher Scientific) for 1 h, at room temperature, washed once in TBS-T and incubated with the respective antibodies for 2 h at room temperature or overnight at 4°C. After three washes with TBS-T, the membranes were incubated with the respective HRP-conjugated secondary antibody in TBS-T for

1 h at room temperature, washed three times in TBS-T, and imaged on an iBright Imaging system (Thermo Fisher Scientific) using Immobilon Forte Western HRP substrate (Merck Millipore). Recombinant FcStrep, gH-FcStrep or gH-FcStrep/gL-Flag complexes were precipitated from the supernatant of 293T cells transfected with the respective expression constructs. Supernatant was collected 2 d after transfection, filtered through 0.45 μm membranes (Millipore) and incubated with magnetic StrepTactinXT beads (IBA Lifesciences, Saint Louis, MO) overnight at 4°C with agitation. After three washes in NP40 lysis buffer, StrepTactinXT beads with pre-coupled FcStrep, gH-FcStrep or gH-FcStrep/gL-Flag complexes were incubated overnight at 4°C with agitation with myc-tagged Eph receptor containing lysates normalized for Eph receptor expression levels as determined by immunoblot. Volumes were adjusted with lysate of untransfected 293T cells. After overnight incubation, magnetic StrepTactinXT beads were washed three times in NP40 lysis buffer on Dynamag magnetic racks. Precipitates were heated in 1x LDS sample buffer (70°C, 10 min) and immunoblot was performed as described above (see S1 Table for a complete list of antibodies).

## Enzyme-linked immunosorbent assay

To measure binding of soluble Eph proteins or MHV68 gH/gL-specific antibodies in mouse serum, clear flat-bottom 96-well Maxisorp plates (Thermo Fisher Scientific, #439454) were coated with recombinant MHV68 gL-gH fusion protein at 2.5 μg/ml in PBS overnight at room temperature. After three washes in 0.05% Tween20 in PBS (PBS-T), wells were blocked with 1% BSA in PBS (R&D Systems #DY995) for 1 h at room temperature. Incubation with soluble Eph proteins at the indicated concentration or serum at the indicated dilutions was performed for 2 h at room temperature in 1% BSA in PBS. After three washes in PBS-T, bound protein was detected using horseradish peroxidase-conjugated ProteinG (Merck Milipore #18–161) at 12.5 ng/ml (soluble Eph proteins) or goat-anti-mouse IgG (Invitrogen #31430) (serum) at 10 ng/ml in in 1% BSA in PBS. After three washes, a 1:1 mixture of Color Reagent A ($H_2O_2$) and Color Reagent B (Tetramethylbenzidine) (R&D Systems) was added. The reaction was stopped by adding 2 N $H_2SO$ (R&D Systems). Absorbance at 450 nm and 540 nm (reference wavelength) was read on a Biotek Synergy 2 plate reader (Agilent, Santa Clara, CA). Correction for non-specific absorbance was per performed by subtracting absorbance at 540 nm and blank controls from absorbance at 450 nm.

## Infection assays and flow cytometry

For infection assays of adherent cell lines, cells were plated at 12,500 cells per well in 96 well plates. One day after plating, cells were infected with the indicated amounts of virus. Block of MHV68 infection with soluble decoy receptor was assayed by infection with MHV68 ORF59-GFP inocula that were pre-incubated with the indicated concentrations of soluble mEphA2-Fc, mEphA4-Fc, mEphA6-Fc or mEphB3-Fc at room temperature for 30 min. 16 h post infection (hpi) cells were harvested by brief trypsinization, followed by addition of 5% FCS in PBS to inhibit trypsin activity, spun down (1,500 rpm, 5 min), washed once with PBS, re-pelleted, fixed in PBS supplemented with 2% formaldehyde (ThermoFisher Scientific) and analyzed for GFP expression on a Cytoflex flow cytometer (Beckman Coulter, Brea, CA). Transduced Raji cells were plated at 25,000 cells per well in 96 well plates, virus was added at MOI 10–30, followed by centrifugation (1,500 × g, 1 h, RT). Micrographs of GFP⁺ Raji cells were taken at 24 hpi and cells were harvested for flow cytometry. Data analysis was performed in FlowJo (v10). Raw data was log-transformed and normalized to controls prior to statistical analysis and back transformed for visualization.

## MHV68 neutralization and antibody depletion assays

For neutralization assays, murine serum was serially diluted and incubated with MHV68 ORF59-GFP for 30 min at room temperature. Subsequently, serum/virus inocula were added to cells plated at 12,500 cells in 96 well plates one day prior to infection. 16 h post infection cells were harvested by brief trypsinization, followed by addition of 5% FCS in PBS to inhibit trypsin activity, spun down (1,500 rpm, 5 min), washed once with PBS, re-pelleted and fixed in PBS supplemented

with 2% formaldehyde and analyzed for GFP expression on a Cytoflex flow cytometer. Data was analyzed using FlowJo (v10). For depletion of MHV68 gH/gL-specific antibodies, recombinant FcStrep, gH$^{ecto}$-FcStrep/gL-Flag or gH$^{D-l}$-FcStrep/gL-Flag complexes were pre-coupled to magnetic StrepTactinXT beads overnight at 4°C with rotation as described above. After three washes in PBS/ 1% FCS/ 2 mM EDTA on a Dynamag magnetic rack StrepTactinXT beads with pre-coupled FcStrep, gH$^{ecto}$-FcStrep/gL-Flag or gH$^{D-l}$-FcStrep/gL-Flag complexes were incubated with murine serum to adsorp gH/gL-binding antibodies for at least 5 h at 4°C with rotation. A total of three adsorption steps were performed. Serum pre- and post-depletion was stored at -80°C. Evaluation of MHV68 neutralizing potential of depleted sera was performed as described above.

## Structure prediction and analysis

The MHV68 gH ectodomain (NP_044860, V25-Q702) was predicted in complex with MHV68 gL (AAF19311, C21-W137) using ColabFold v1.5.5, followed by relaxing of the top ranked structure using amber force fields (pLDDT = 84.1 pTM = 0.865 ipTM = 0.912) [87]. N- and C-terminal residues with pLDDT < 70 were excluded in the visualization. Truncated chains for MHV68 gH (NP_044860.1, residues 1–95) and gL (NP_044884, residues 1–137) were predicted in complex with either murine EphA4 (NP_031962) or murine EphB3 (NP_034273) using ColabFold (NP_031962: mean pLDDT = 60.9, median pLDDT = 61.5; NP_044884: mean pLDDT = 73.3, median pLDDT = 84.6). The resulting models were aligned against known structures (PDBIDs 3PHF, 7CZE, 7B7N, 7CZF) and closest matching MHV68 gH/gL-mEphA4 model (PBDid 7B7N, rmsd = 1.1 Å, DALI Z = 25.5) was refined using one round of Rosetta fast_relax to minimize steric clashes and improve the hydrogen bonding network [88]. Further analysis and visualization was performed in ChimeraX [89].

## Mathematical and statistical analysis

Curve fitting of binding of Eph receptors and gH/gL-directed antibodies from mouse serum to gL-gH fusion protein was performed using a built-in A 4-parameter logistic (4PL) curve in GraphPad Prism version 10 (GraphPad Software, La Jolla, CA). Statistical analysis of multiple groups was performed using ordinary one-way analysis of variance (ANOVA) followed by Holm-Šídák's multiple comparisons test (Figs 2D–2G, 3D) or ordinary two-way ANOVA followed by Tukey's multiple comparisons test (Figs 5E, S4A). All Statistical analyses were performed with GraphPad Prism version 10. For all statistics, *: p-value < 0.05, **: p-value < 0.01, ***: p-value < 0.001, ****: p-value < 0.0001.

## Supporting information

**S1 Fig. The MHV68 gH/gL complex binds human and murine Eph receptors.** (A) Pairwise precipitation of soluble recombinant FcStrep with individual human Eph proteins. Precipitates were analyzed by immunoblot with indicated antibodies. Asterisks indicate known KSHV, EBV or RRV gH/gL interaction partners. (B) Input immunoblot for individual human Eph proteins. (C) Schematic representation of alternative open reading frames of MHV68 gL. Alternative start codons are indicated by arrowheads, SP: predicted signal peptide. Pairwise precipitation of soluble recombinant MHV68 gH ectodomain in complex with expression constructs for full-length MHV68 gL (gH/gL) or MHV68 gL$^{\Delta N36}$ (37–173) (based on NP_044884.3) with murine EphA4. MHV68 gH alone was used as control. Precipitates were analyzed by immunoblot with indicated antibodies. (D) Protein accession numbers and percentage of identical amino acids in aligned regions of human and murine Eph proteins as determined by BLAST (Basic Local Alignment Search Tool). (E) Pairwise precipitation of soluble recombinant gH-FcStrep with individual murine Eph proteins. Precipitates were analyzed by immunoblot with indicated antibodies. For A, B, C, E, molecular weight is indicated in kDa.
(TIF)

**S2 Fig. Soluble murine EphA4 and EphB3 inhibit MHV68 infection of endothelial cells and fibroblasts.** (A-D) Cell type-dependent inhibition of MHV68 infection by soluble murine Eph proteins at 100 nM homodimerized protein. EphA2-Fc

and PBS were used as controls. GFP expression as indicator of infection was measured by flow cytometry. Infection is shown as percentage of GFP$^+$ cells. Symbols representing individual experiments are shown; lines connect values from the same experiment.
(TIF)

**S3 Fig. Multiple residues on gH and gL mediate the MVH68 gH/gL Eph interaction.** (A) Multiple sequence alignment of gH D-I of human (KSHV [QFU18817], EBV [AIM62235]), rhesus macaque (RRV 26–95 [AAF60000]) and murine (MHV68 [NP044860]) GHVs. Numbers according to MHV68 gH. Arrowhead indicates R106$^{EphA4/EphB3}$ interacting residue D52$^{MHV68\ gH}$. (B) Single amino acids in the putative Eph interacting region in MHV68 gH were mutated to alanine. MHV68 gH$^{ecto}$ mutants in complex with MHV68 gL were precipitated with murine EphA4. MHV68 gH-FcStrep was used as control. Precipitates were analyzed by immunoblot with indicated antibodies. (C) Putative Eph-interacting residues in MHV68 gL were mutated to alanine. MHV68 gH$^{ecto}$ in complex with MHV68 gL mutants was precipitation with murine EphA4. MHV68 gH-FcStrep was used as control. Precipitates were analyzed by immunoblot with indicated antibodies. For B, C, molecular weight is indicated in kDa.
(TIF)

**S4 Fig. Neutralizing antibodies in MHV68 infected mice target the MHV68 gH/gL complex.** (A) Antibodies to gH$^{ecto}$/gL or gH$^{D-I}$/gL were depleted using soluble complexes pre-coupled to magnetic beads. Fc was used as control. gH/gL-specific IgG from naïve or MHV68-infected C57BL/6 before and after adsorption was measured by MHV68 gL-gH ELISA. Background corrected optical density at 450 nm is shown. Mean and symbols representing individual experiments are shown. (B) Correlation of mean neutralization and optic density from three independent experiments. (C) Serum neutralization of MHV68 ORF59-GFP infection on NIH 3T3 cells is mediated by gH/gL-targeting antibodies. Antibodies to gH$^{ecto}$/gL or gH$^{D-I}$/gL were depleted using soluble complexes pre-coupled to magnetic beads. Fc was used as control. Micrographs were taken at 16 hpi. Statistical significance was evaluated by ordinary two-way ANOVA followed by Tukey's multiple comparisons test. ***: p-value < 0.001, ****: p-value < 0.0001.
(TIF)

**S1 Table. List of primers and antibodies used in the study.**
(XLSX)

**S2 Table. Primary infection data for** Figs 2, 3 and 5.
(XLSX)

## Acknowledgments

We thank Morgan Pagonis, Nick Wright, Britney Erickson, Ashley Mitchell, Kayla Russell, Vanessa Wall, and Jane Jones from the Protein Expression Laboratory at the Frederick National Laboratory for cloning and protein expression support. We thank Alexander Hahn for the generous sharing of plasmids. We also thank Amber Zeng for technical support and members of the Krug laboratory for helpful discussion.

## Author contributions

**Conceptualization:** Anna K. Großkopf, Laurie T. Krug.

**Data curation:** Anna K. Großkopf.

**Formal analysis:** Anna K. Großkopf, Victor Tobiasson.

**Funding acquisition:** Anna K. Großkopf, Laurie T. Krug.

**Investigation:** Anna K. Großkopf, Victor Tobiasson.

**Methodology:** Anna K. Großkopf.

**Project administration:** Anna K. Großkopf, Laurie T. Krug.

**Resources:** Laurie T. Krug.

**Supervision:** Laurie T. Krug.

**Validation:** Anna K. Großkopf.

**Visualization:** Anna K. Großkopf, Victor Tobiasson.

**Writing – original draft:** Anna K. Großkopf.

**Writing – review & editing:** Anna K. Großkopf, Victor Tobiasson, Laurie T. Krug.

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
