## [Decision Letter · Decision Letter 0]

10 Jul 2025

PPATHOGENS-D-25-01356

Eph Receptor Tyrosine Kinases Are Functional Entry Receptors for Murine Gammaherpesvirus 68

PLOS Pathogens

Dear Dr. Großkopf,

Thank you for submitting your manuscript to PLOS Pathogens. After careful consideration, we feel that it has merit but does not fully meet PLOS Pathogens's publication criteria as it currently stands. Therefore, we invite you to submit a revised version of the manuscript that addresses the points raised during the review process.

Please submit your revised manuscript within 30 days Sep 08 2025 11:59PM. If you will need more time than this to complete your revisions, please reply to this message or contact the journal office at plospathogens@plos.org. Please include the following items when submitting your revised manuscript:

We look forward to receiving your revised manuscript.

Kind regards,

Kenneth M Kaye

Academic Editor

PLOS Pathogens

Robert Kalejta

Section Editor

PLOS Pathogens

Sumita Bhaduri-McIntosh

Editor-in-Chief

PLOS Pathogens

orcid.org/0000-0003-2946-9497

Michael Malim

Editor-in-Chief

PLOS Pathogens

orcid.org/0000-0002-7699-2064

**Additional Editor Comments:**

Three reviewers have reviewed your manuscript and all were very enthusiastic about the findings and presentation. However, there are a few minor issues that were raised that should be addressed.

**Journal Requirements:**

At this stage, the following Authors/Authors require contributions: Anna Katharina Großkopf, Victor Tobiasson, and Laurie T. Krug. Please ensure that the full contributions of each author are acknowledged in the "Add/Edit/Remove Authors" section of our submission form.

2) We notice that your supplementary Figures are included in the manuscript file. Please remove them and upload them with the file type 'Supporting Information'. Please ensure that each Supporting Information file has a legend listed in the manuscript after the references list.

3) Some material included in your submission may be copyrighted. According to PLOSu2019s copyright policy, authors who use figures or other material (e.g., graphics, clipart, maps) from another author or copyright holder must demonstrate or obtain permission to publish this material under the Creative Commons Attribution 4.0 International (CC BY 4.0) License used by PLOS journals. Please closely review the details of PLOSu2019s copyright requirements here: PLOS Licenses and Copyright. If you need to request permissions from a copyright holder, you may use PLOS's Copyright Content Permission form.

Potential Copyright Issues:

- Figure 5A. Please confirm whether you drew the images / clip-art within the figure panels by hand. If you did not draw the images, please provide (a) a link to the source of the images or icons and their license / terms of use; or (b) written permission from the copyright holder to publish the images or icons under our CC BY 4.0 license. Alternatively, you may replace the images with open source alternatives. See these open source resources you may use to replace images / clip-art:

4) Please ensure that the funders and grant numbers match between the Financial Disclosure field and the Funding Information tab in your submission form. Note that the funders must be provided in the same order in both places as well.

State what role the funders took in the study. If the funders had no role in your study, please state: "The funders had no role in study design, data collection and analysis, decision to publish, or preparation of the manuscript.".

**Reviewers' Comments:**

Reviewer's Responses to Questions

**Part I - Summary**

Reviewer #1: Grosskopf et al report here that the Eph receptor tyrosine kinases function as receptors for murine gammaherpevirus 68. The EphR TKs were previously shown to serve as receptors for EBV, KSHV and RRV through binding the conserved gH/gL glycoproteins, findings that underpin the significance of this new report. Vaccination and immune protection from gammaherpesvirus infection and subsequent diseases is an important goal, and therefore demonstration that the mouse model system displays remarkable conservation of the critical entry interaction is very important to the field and to future studies.

This report is thorough and straightforward, making use of in silico comparative studies, screening of multiple EphR family members, use of purified viral glycoprotein complexes, analysis of MHV68 infection and neutralization with purified complexes and domains, and demonstration of the sufficiency of the EphR A4 and B3 for infection of human cells. The conclusions of this report are fully supported, and indicate the utility of the conserved MHV68 infection entry system for modeling and testing gammaherpesvirus entry.

Reviewer #2: Prior research has identified Eph receptors as key entry receptors engaged by the gH/gL complex of human gammaherpesviruses. This study by Grosskopf et al. is the first to demonstrate Eph receptors, specifically EphA4 and EphB3, as functional entry receptors for murine gammaherpesvirus 68 (MHV68). The manuscript presents multiple lines of evidence supporting their conclusion: interactions of MHV68 gH/gL with EphA4 and EphB3, inhibition of infection in permissive cell types by soluble EphA4 and B3, and the ability of ectopically expressed EphA4 and B3 to render otherwise non-permissive human Raji B cells susceptible to MHV68 infection. These experiments include another Eph receptor, EphA2 as a negative control for specificity. Guided by modeling and alignment with the existing structure of KSHV gH/gL-EphA2 complex, the authors experimentally demonstrated that the interaction between MHV68 gH/gL and EphA4 involves specific residues at domain I of MHV68 gH. This confirms evolutionary conservation of entry mechanisms. Overall, this study represents a significant advancement in gammaherpesvirus research and establishes a strong foundation for using MHV68 as an in vivo model system for pathogenesis and vaccine development. The manuscript is well written, clearly presented with solid experimental data, and easy to follow. Only a few clarifications are needed to further enhance the clarify of the study.

Reviewer #3: Although entry receptors for EBV and KSHV, including Eph family members, have been well-described, entry receptors for the related murine virus MHV68 have not been previously determined. Work in this manuscript directly tests the ability Eph family members to interact with MHV68 glycoproteins and serve as entry receptors. Through a series of beautiful experiments, the authors identify EphA4 and EphB3 as direct interaction partners of the MHV68 gH/gL complex, and further demonstrate that expression of EphA4 and B3 facilitates infection of multiple cell types from both humans and mice. This is outstanding, well-written report with carefully controlled experiments and clear conclusions. These exciting results provide a large step forward for the field and provide important additional evidence for the evolutionary conservation of MHV68 with other human and primate herpesviruses.

**Part II – Major Issues: Key Experiments Required for Acceptance**

Reviewer #1: no major issues

Reviewer #2: None

Reviewer #3: NONE

**Part III – Minor Issues: Editorial and Data Presentation Modifications**

Reviewer #1: Minor issues that could be discussed:

1) In figure 2, it appears that soluble EphA2 trends toward enhancement of infection. It would be informative to learn whether this is considered to be technical or specific.

2) As shown in figure 5, antisera depletion of gH/gL specific antibodies does decrease neutralization of infection, but only to approximately 50% as the authors point out. Is this due to the capacity for herpesviruses to use multiple means of entry, incomplete depletion of gH/gL antibodies specific to epitopes not shared with the purified gH/gL reagent, other? Some discussion of this is useful in thoughtful consideration of focusing on neutralizing gH/gL in gammaherpesvirus vaccination.

Reviewer #2: 1. Fig. 2. Are all cell types equally susceptible to MHV68 infection? In other words, what are the percentage of GFP+ cells in the PBS controls? This information should be included.

2. LINE 180: Typically, lower EC50 values indicates higher affinity. Therefore, the higher EC50 observed for mEphA4 is somewhat counterintuitive. The authors should clarify the interpretation of EC50 and maximum OD values in the context of receptor and gH/gL interactions.

3. Fig. 3D. Only fold change is displayed. The authors should indicate what a >20-fold increases correspond to regarding the percentage of GFP+ cells.

4. Fig. 5B. The authors should indicate whether undiluted or diluted serum was used. If diluted serum was used for ELISA, include the dilution factor.

5. Does KSHV gH/gL complex interact with murine EpHA2 and A4?

Reviewer #3: MINOR

1. Fig 2C – 2G. Figures indicate percent GFP relative to control, but give no indication of total GFP+ cells (ie, percent cells infected). This should be stated clearly (or raw data included), perhaps as a supplementary figure or table.

2. MHV68 ORF59-GFP should be described in slightly more detail in the main text to indicate the nature of the GFP expression and the role of ORF59.

PLOS authors have the option to publish the peer review history of their article (what does this mean? ). If published, this will include your full peer review and any attached files.

**Do you want your identity to be public for this peer review?** For information about this choice, including consent withdrawal, please see our Privacy Policy .

Reviewer #1: No

Reviewer #2: No

Reviewer #3: No

**Figure resubmission:**
---

## [Decision Letter · Decision Letter 1]

6 Oct 2025

Dear Dr. Großkopf,

We are pleased to inform you that your manuscript 'Eph Receptor Tyrosine Kinases Are Functional Entry Receptors for Murine Gammaherpesvirus 68' has been provisionally accepted for publication in PLOS Pathogens.

Best regards,

Kenneth M Kaye

Academic Editor

PLOS Pathogens

Robert Kalejta

Section Editor

PLOS Pathogens

Sumita Bhaduri-McIntosh

Editor-in-Chief

PLOS Pathogens

orcid.org/0000-0003-2946-9497

Michael Malim

Editor-in-Chief

PLOS Pathogens

orcid.org/0000-0002-7699-2064

Reviewer Comments (if any, and for reference):

Reviewer's Responses to Questions

**Part I - Summary**

Reviewer #1: Grosskopf et al report here that the Eph receptor tyrosine kinases function as receptors for murine gammaherpevirus 68. The EphR TKs were previously shown to serve as receptors for EBV, KSHV and RRV through binding the conserved gH/gL glycoproteins, findings that underpin the significance of this new report. Vaccination and immune protection from gammaherpesvirus infection and subsequent diseases is an important goal, and therefore demonstration that the mouse model system displays remarkable conservation of the critical entry interaction is very important to the field and to future studies.

This report is thorough and straightforward, making use of in silico comparative studies, screening of multiple EphR family members, use of purified viral glycoprotein complexes, analysis of MHV68 infection and neutralization with purified complexes and domains, and demonstration of the sufficiency of the EphR A4 and B3 for infection of human cells. The conclusions of this report are fully supported, and indicate the utility of the conserved MHV68 infection entry system for modeling and testing gammaherpesvirus entry.

Reviewer #2: The authors have thoughtfully addressed the comments raised by the reviewers. I have no further concerns. Overall, this is a well-executed study.

Reviewer #3: Reviewers have appropriately addressed all concerns.

**Part II – Major Issues: Key Experiments Required for Acceptance**

Reviewer #1: none

Reviewer #2: (No Response)

Reviewer #3: (No Response)

**Part III – Minor Issues: Editorial and Data Presentation Modifications**

Reviewer #1: Inclusion of the raw counts for %GFP+ cells is appreciated. While looking closely at the raw data makes it clear why the authors chose to display as fold-increase vs PBS, and those insights are supported by the raw data, it may be more transparent to actually name the variability in the raw data. The raw data baselines are highly variable, and for readers who would replicate findings, it would be helpful to make it clear that the %GFP cells across experiments has wide variability. This does not diminish the results, but would enhance transparency.

Reviewer #2: (No Response)

Reviewer #3: (No Response)

PLOS authors have the option to publish the peer review history of their article (what does this mean? ). If published, this will include your full peer review and any attached files.

**Do you want your identity to be public for this peer review?** For information about this choice, including consent withdrawal, please see our Privacy Policy .

Reviewer #1: No

Reviewer #2: No

Reviewer #3: No

---

## [Editor Report · Acceptance letter]

Dear Dr. Großkopf,

We are delighted to inform you that your manuscript, "Eph Receptor Tyrosine Kinases Are Functional Entry Receptors for Murine Gammaherpesvirus 68," has been formally accepted for publication in PLOS Pathogens.

Best regards,

Sumita Bhaduri-McIntosh

Editor-in-Chief

PLOS Pathogens

orcid.org/0000-0003-2946-9497

Michael Malim

Editor-in-Chief

PLOS Pathogens

orcid.org/0000-0002-7699-2064